# Evaluation of Membrane Fouling Control for Brackish Water Treatment Using a Modified Polyamide Composite Nanofiltration Membrane

**DOI:** 10.3390/membranes13010038

**Published:** 2022-12-28

**Authors:** Xuebai Guo, Cuixia Liu, Bin Feng, Yuanfeng Hao

**Affiliations:** 1Department of Environmental Engineering, Henan Vocational College of Water Conservancy and Environment, Zhengzhou 450008, China; 2School of Energy & Environment, Zhongyuan University of Technology, Zhengzhou 450007, China; 3CCTEG Chongqing Engineering (Group) Co., Ltd., Chongqing 401331, China

**Keywords:** modified nanofiltration membrane, membrane fouling, brackish water, operation condition, cleaning procedure

## Abstract

In northwest China, the limited amount of water resources are classified mostly as brackish water. Nanofiltration is a widely applied desalination technology used for brackish water treatment; however, membrane fouling restricts its application. Herein, we modified the membrane with triethanolamine (TEOA) and optimized the operating conditions (transmembrane pressure, temperature, and crossflow velocity) to control the nanofiltration membrane fouling by brackish water. Based on the physiochemical characteristics and desalination performance of the prepared membranes, the membrane modified with 2% TEOA (MPCM2) was identified as the optimal membrane, and 0.5 MPa, 25 °C, and 7 cm/s were identified as the optimal operating conditions through a series of nanofiltration experiments. Moreover, the membrane cleaning procedure for fouled MPCM2 was further determined, and a two-step cleaning procedure using ethylene diamine tetraacetic acid disodium followed by HCl with a permeance recovery rate of 98.77% was identified as the optimal cleaning procedure. Furthermore, the characterizations of the fouled and cleaned MPCM2 showed that the optimized cleaning procedure could recover the properties of MPCM2 to near virgin. This study is of great significance for the long-term stable operation of nanofiltration processes in brackish water treatment to ensure the supply of healthy water in the water-deficient areas of northwest China.

## 1. Introduction

With the development of the global economy and the rapid growth of the global population, the shortage of water resources has become a concerning issue worldwide [1,2]. Although China has abundant water resources, its water resources per capita are far below the global average [3,4]. Moreover, water resources in China are not evenly distributed among the different regions [5,6]. Many regions lack water resources, especially the northwest regions, where the limited water resources are mostly brackish water [7,8]. Brackish water, characterized by total dissolved solids (TDS) of 500–35,000 mg/L, cannot be directly consumed [9]. The long-term use of brackish water is harmful to human health, and the various salt ions in brackish water may also be harmful to industrial and agricultural production [10]. Therefore, it is necessary to develop a rapid treatment process with low operating costs to treat brackish water for consumption.

Desalination is essentially the process of separating salts from water. Advances in desalination technologies have made significant progress over the past 50 years [11,12]. Existing desalination technologies include distillation [13,14], freezing [15,16], chemical precipitation [17], electroosmosis [12,18], and membrane separation technologies such as reverse osmosis (RO) [19,20] and nanofiltration [7,21,22]. Among these technologies, membrane separation is a highly promising technology due to its high separation efficiency. Though efficient, RO requires a high operating pressure and consumes considerable energy [20,23,24]. In contrast, the operating pressure for nanofiltration is much lower, and with its small footprint, nanofiltration can be easily achieved with automatic control, making it suitable for use in many fields [25,26,27,28].

Membrane fouling decreases membrane permeance, increases cleaning frequency, shortens the lifespan of the membrane, and increases operating costs [29,30,31,32]. Factors affecting nanofiltration membrane fouling can be generally grouped into three aspects [33,34,35]: the feed characteristics such as the pH and ionic conditions; the membrane characteristics such as roughness and hydrophilicity; and the operating conditions such as transmembrane pressure (TMP), temperature, crossflow velocity, and the structure of the membrane module. These factors directly affect membrane fouling, and a suitable optimization of these factors can help to control membrane fouling [36,37]. On the other hand, cleaning is commonly required if the membrane permeance decreases to a certain degree, with the desire to restore the nanofiltration performance. Therefore, research on the mitigation of membrane fouling or optimization of cleaning procedures is necessary for the further development of nanofiltration technology. Notably, most of the literature has only focused on one direction of membrane modification, the optimization of the operating conditions, or majorization of the cleaning procedure to control membrane fouling. A comprehensive study that considers the three aspects of membrane modification, the optimization of operating conditions, and the majorization of the cleaning procedure is urgently needed. Based on the real-world setting of water resources in northwest China, this study aimed to control the nanofiltration membrane fouling by brackish water through membrane modification and optimizing the operating and cleaning conditions, which are of great significance to ensure the long-term and stable operation of a nanofiltration system and healthy water in water shortage areas.

In this study, first, a polyamide composite membrane (PCM) was prepared based on an interfacial polymerization process, and its performance was optimized by testing the pure water contact angle (CA), zeta potential (ZP), carboxylic acid group (CAG) density, ultrapure water permeance, and desalination efficiencies of Na_2_SO_4_ and MgCl_2_. Second, the operating conditions including TMP, temperature, and crossflow velocity of the nanofiltration process were optimized based on the permeance attenuation behavior and removal efficiency of brackish water. Third, a membrane fouling analysis for the MPCM under optimal operating conditions was conducted using X-ray photoelectron spectroscopy (XPS) and atomic force microscopy (AFM), and four classical membrane fouling models were utilized for the fitting of membrane permeance to illustrate the membrane fouling mechanism. Finally, the cleaning procedure was optimized, and the influence of membrane cleaning on the membrane properties was discussed. This study is of great significance for the long-term stable operation of nanofiltration in brackish water to ensure the consumption of healthy water in water-deficient areas of northwest China.

## 2. Materials and Methods

### 2.1. Materials

The polysulfone (PS) membranes used in this study have stable chemical properties, robust mechanical strength, resistance to acid and alkali corrosion [38]. The PS membrane, with a molecular weight cut-off of 30 kDa, was produced by Solecta, Inc. (Oceanside, CA, USA). Sodium dodecyl sulfate (SDS) and NaOH were purchased from Tianjin Benchmark Chemical Reagents Co., Ltd. (Tianjin, China). Ethylene diamine tetraacetic acid disodium (EDTA) was supplied by Tianjin Kemiou Chemical Reagent Co., Ltd. (Tianjin, China). Piperazine (PIP), trimesoyl chloride (TMC), triethylamine (TEA), and polyethylene glycol 200 (PEG200) were purchased from Shanghai Aladdin Reagent Co., Ltd. (Shanghai, China). N-hexane and triethanolamine (TEOA) were purchased from Sinopharm Chemical Reagent Co., Ltd. (Shanghai, China). All reagents were of analytical grade.

The brackish water used in this study was obtained from northwest China. Before the nanofiltration desalination process, the brackish water was treated using a conventional surface water treatment process, including coagulation, sedimentation, and quartz sand filtration. The outlet water of the quartz sand filtration process was the inlet water for the nanofiltration. Appendix A lists the water quality parameters of the nanofiltration inlet water.

### 2.2. Preparation of PCMs and MPCMs

An interfacial polymerization process was used to prepare the PCMs with PIP and TMC [39]. The specific preparation method is as follows. 

Step 1: The PS membrane was placed in a 1% ethanol solution at 30 °C and mixed for 4 h. Next, the membrane was placed in deionized water, sonicated for 1 min, and stored in ultrapure water. Step 2: The cleaned membrane was immobilized on a glass plate, and 50 mL of 0.5% PIP water solution (pH = 11, adjusted with TEA) was poured onto the membrane surface, which was poured away after 3 min, and the excess water on the membrane surface was removed using a degreasing cotton. Step 3: A solution of 50 mL of 0.1% TMC n-hexane was poured onto the membrane surface, reacted for 30 s, and the membrane surface was immediately rinsed with n-hexane to remove the unreacted TMC molecules. Step 4: The membrane was placed in a drying oven at 50 °C for 15 min. Step 5: A solution of 0.5, 1, 2, or 3% TEOA (pH = 11, adjusted with TEA) was poured onto the membrane surface and reacted for 5 min. Step 6: The membrane was placed in deionized water, vibrated for 4 h to remove unreacted monomers, and stored in ultrapure water at 4 °C.

The membranes prepared without step 5 were called PCMs, and the MPCMs prepared with 0.5, 1, 2, and 3% TEOA water solutions were called MPCM0.5, MPCM1, MPCM2, and MPCM3, respectively.

### 2.3. Nanofiltration Experiments

Nanofiltration experiments were performed using a continuous flow nanofiltration set-up (Figure 1). The prepared nanofiltration membrane, with an effective area of 0.005 m^2^, was installed in the membrane unit, and the plunger pump under the transducer provided less than 1.0 MPa operating pressure for the nanofiltration process. The effluent of the membrane unit was collected in a beaker and weighed on an electronic scale on a real-time basis. The feed line by path and concentrated water were routed back into the feed tank, and a temperature controller was used to maintain a constant temperature. The crossflow velocity on the membrane surface was controlled by adjusting the reflux valve of the concentrated water.

Each prepared membrane was first used to filter deionized water at 25 °C and 1.0 MPa until the permeance was consistent. Then, deionized water was replaced with 40 L of pretreated brackish water, the plunger pump was opened when the temperature reached the preset temperature for 1 h, and the TMP and crossflow velocity were adjusted to the set value; thereafter, the nanofiltration experiment began. The nanofiltration pipeline was cleaned after each experiment for later use. In order to optimize the operation conditions of nanofiltration, TMP, temperature, and crossflow velocity were optimized step by step. A gradient of TMP was set to 0.3, 0.5, 0.7, and 0.9 MPa [40]. A gradient of temperature was set to 20, 25, and 30 °C [41]. A gradient of crossflow velocity was set to 4, 7, and 10 cm/s [42].

### 2.4. Membrane Fouling Models

Membrane fouling models can be used for the elaboration of membrane fouling, and four classical membrane fouling models were used to fit the membrane permeance decline at constant pressure. The four models are the complete blocking, standard blocking, intermediate blocking, and cake filtration models [43], whose expressions are listed in Appendix A. The fitting parameters of the correlation coefficient (R^2^) and the sum of squared errors (SSE) are used to estimate the fitting degree of the membrane fouling models with membrane permeance.

### 2.5. Membrane Cleaning Experiments

The fouled membranes used for membrane cleaning experiments were obtained at the optimum TMP, temperature, and crossflow velocity. Moreover, the nanofiltration process was stopped when the membrane permeance was attenuated to 80% of the initial membrane permeance, and fouled membranes were obtained. Different cleaning agents were placed in the feed tank to replace the brackish water before cleaning, and the piston pump was opened for 30 min until the temperature of the cleaning agents in the feed tank stabilized at 25 °C. The reflux valve was adjusted to enhance the crossflow velocity on the membrane surface to 15.0 cm/s, and the TMP was set to 0 MPa. Deionized water was used to replace the cleaning reagent, and the cleaning steps were repeated.

Permeance recovery rate (*PR*, %) is calculated as follows:(1)$$PR=\frac{J_{c}-J_{f}}{J_{0}-J_{f}}$$
where *J_0_*, *J_f_*, and *J_c_* represent the pure water permeances of the original, fouled, and cleaned MPCMs, respectively, at 25 °C with TMP of 0.8 MPa and cross flow velocity of 7.0 cm/s.

### 2.6. Characterizations of Membrane and Water Quality

The CA and ZP values of the membrane surface were measured using a contact angle tester (SL200B3, Shanghai, China) and a solid ZETA potential analyzer (SurPASS, Antompa, Austria), respectively. Three parallel tests for the measurements of the CA and ZP values were performed, and the average values were taken as the test values. The CAG density of the membrane surface was measured using toluidine blue adsorption [44]. XPS (PHI 5700 ESCA, Chanhassen, MN, USA) was used to determine the elemental composition of the fouled membranes and AFM (Bioscope, Hamtramck, MI, USA) was used to observe the morphology of the membrane surface.

The concentration of positive ions was measured using inductively coupled plasma-atomic emission spectrometry (Plasma1000, US). The solution pH was measured using a multiparameter analyzer (DZS-706, China). Total hardness, total dissolved solids (TDS), and chemical oxygen demand (COD) were measured using the sanitary standard inspection method for drinking water (GBT5750-2006) in China. 

## 3. Results and Discussion

### 3.1. Preparation and Selection of the MPCMs

#### 3.1.1. CA, ZP, and CAG Density

Figure 2 shows the CA, ZP, and CAG density of the prepared membranes. The PCM had the lowest CA, ZP, and CAG density, suggesting that modification with TEOA had a significant influence on the membrane surface properties. The CA, ZP, and CAG density of the PCMs were 41.42°, –57 mV, and 86 CAGs per nm^2^, respectively. The modifications with 0.5, 1, 2, and 3% TEOA decreased the values of the CA from 41.42 to 37.33, 30.62, 27.58, and 27.15°, increased ZP from −57 to −51, −46, −41, and −39 mV, and decreased the CAG density from 86 to 71, 58, 46, and 43 CAGs per nm^2^, respectively. All three indicators significantly varied with the increasing TEOA content from 0.5 to 2%, whereas further enhancement in the three indicators became less significant with a further increase in TEOA content.

The decrease in CA values suggests that the MPCM hydrophilicity was enhanced to some extent, owing to the appearance of abundant oxhydryl groups on the membrane surface. The unmodified PCM with abundant carboxyl groups was hydrophilic, whereas the two oxhydryl groups obtained as a carboxyl group were consumed during the modification process, which enhanced its hydrophilicity. The formation of a hydrated layer was inevitable because of the combination of oxhydryl groups on the MPCM surface with water molecules. Therefore, the CA of MPCMs was lower than that of PCM, and the MPCMs were more hydrophilic, which could mitigate membrane fouling. This was also the reason for the lower negative charge density (higher ZP values) and CAG density of the MPCMs.

#### 3.1.2. Pure Water Permeance and Desalination Performance

Figure 3 shows the pure water permeance and desalination efficiencies (Na_2_SO_4_ and MgCl_2_) of the PCMs and MPCMs at 25 °C with a TMP of 0.5 MPa. The modification with TEOA enhanced the pure water permeance and removal efficiency of MgCl_2_ to a certain extent and reduced the removal efficiency of Na_2_SO_4_. The enhancement in the pure water permeance was due to the increase in the hydrophilicity of the MPCMs, which allowed water molecules to more easily pass through the MPCMs. Chen et al. [45] reported that stronger electronegativity was favorable for the retention of divalent anion salts (such as Na_2_SO_4_), which was in agreement with the results of our study. Compared to the PCMs, the reduced density of the negative charge as indicated by the ZP value of the MPCMs caused a lower removal efficiency of Na_2_SO_4_ by the MPCMs (Figure 2). The interception of MgCl_2_ by MPCMs may also be related to the lower electronegativity of the MPCMs. MPCM2 was chosen for further research by considering the physiochemical properties (Figure 2), desalination performance (Figure 3), and its enhancement on the permeance attenuation properties of brackish water (Appendix A).

### 3.2. Influence of Operating Conditions on the Nanofiltration of Brackish Water by MPCM2

#### 3.2.1. Influence of TMP on Nanofiltration Performance

##### Permeance Decline Behavior

Figure 4 shows the influence of the operating conditions on the nanofiltration performance of MPCM2, and Appendix A shows the volume ratio of permeate to feed flow (Y) at different operating conditions at the end of nanofiltration. The permeance attenuation curve of MPCM2 for pretreated brackish water with different TMP conditions is shown in Figure 4a, and the percentages of membrane permeance to initial permeance at the end of filtration (permeance attenuation rates) are shown in Figure 4b. With the increase in TMP from 0.3 to 0.9 MPa, the initial permeance of MPCM2 for the pretreated brackish water gradually increased from 50.32 to 134.33 L/(m^2^·h). The attenuation of permeance at 0.3 and 0.5 MPa were much lower than that, at 0.7 and 0.9 MPa, the permeance attenuation rates at 0.3, 0.5, 0.7, and 0.9 MPa were 15.16, 14.01, 26.07, and 32.23%, respectively. The higher initial permeance was due to the larger impetus provided by the higher TMP, which caused more water molecules to pass through the membrane. However, more foulants are rejected and gradually accumulate on the membrane surface at higher TMP conditions, forming a fouling layer, increasing the fouling resistance, intensifying the concentration polarization, and decreasing the membrane permeance [46]. Moreover, the higher TMP caused the higher water production rate, the Y value at the end of nanofiltration increased by 16.26% as the TMP increased from 0.3 to 0.5 MPa, and it increased by only 8.70 and 17.70% as the TMP further increased to 0.7 and 0.9 MPa, respectively (Appendix A).

##### Solute Retention

After 2 h of continuous operation of the nanofiltration system, the permeate solution obtained was used to explore the solute retention efficiency of MPCM2. The removal efficiencies of COD, TDS, and total hardness by the MPCM2 with different TMP conditions are shown in Figure 4b. The COD removal rate for MPCM2 was greater than 91%, while that of TDS and total hardness were higher than 62 and 49%, respectively. With an increase in the TMP, the removal of these indices by MPCM2 improved, suggesting that a higher TMP was beneficial for improving effluent quality. However, the permeance attenuation rate became more significant while the effluent quality was improved. Combining with the properties of membrane permeance attenuation at different TMP conditions in Figure 4a and the Y values in Appendix A, a TMP of 0.5 MPa was selected for further research.

#### 3.2.2. Influence of Temperature on Nanofiltration Performance

##### Permeance Decline Behavior

The permeance attenuation curves at the operating temperatures of 20, 25, and 30 °C are shown in Figure 4c. An increase in temperature elevated the membrane permeance. The initial nanofiltration permeance at 20 °C was 74.81 L/(m^2^·h) and gradually increased to 78.92 and 83.52 L/(m^2^·h) as the temperature increased to 25 and 30 °C, respectively. The increase in temperature slightly decreased the percentage of the final permeance relative to the initial permeance, and the permeance attenuation rate gradually increased with temperature (Figure 4d). Moreover, the Y value at the end of nanofiltration increased from 41.15 to 43.18 and 44.87% by increasing the temperature from 20 to 25 and 30 °C (Appendix A). The aggravation of membrane fouling may be related to the enhancement in the transfer of foulants to the membrane surface by the larger membrane permeance at higher temperatures [47]. Meanwhile, the relatively high temperature was beneficial for the reverse transfer of foulants on the membrane surface to the solution, which is promising for mitigating membrane fouling [48]. Therefore, the experimental results were obtained under the synthetic action of the two effects mentioned above, and the aggravation of membrane fouling was more significant.

##### Solute Retention

Figure 4d shows the removal efficiencies of COD, TDS, and total hardness for MPCM2 at different temperatures. Temperature had different effects on COD, TDS, and total hardness, and an increase in temperature decreased the removal efficiency of COD and increased the removal efficiencies of TDS and total hardness. This difference is strongly related to the different rejection mechanisms of MPCM2 for the various solutes. COD is mainly provided by organic matter, the neutral part of which is primarily rejected by size-exclusion mechanism, whereas TDS and total hardness are mainly provided by salt ions, which are mainly removed through Donnan charge repulsion [49]. As the temperature increases, the nanofiltration membrane pores swell to a certain extent, and the hydrated layer of organic solutes becomes thinner, which causes more neutral organic molecules to pass through MPCM2 and thus a reduced COD removal rate [50], while the electrical characteristics of the nanofiltration membrane are barely affected, including the rejection of salt ions [51]. Therefore, the increase in temperature enhanced the interception of TDS and total hardness by MPCM2.

#### 3.2.3. Influence of Crossflow Velocity on Nanofiltration Performance

##### Permeance Decline Behavior

Figure 4e shows the influence of the crossflow velocity on the membrane permeance, and the crossflow velocities were set to 4, 7, and 10 cm/s. The increase in crossflow velocity had no effect on the initial permeance but substantially affected its attenuation rate. As the crossflow velocity increased from 4 to 7 cm/s, the permeance attenuation rate clearly decelerated, and the percentage of final permeance to initial permeance increased from 70.56 to 85.99%, suggesting that increasing the crossflow velocity was beneficial for controlling membrane fouling. With a further increase in crossflow velocity, the enhancement of the permeance became inconspicuous, resulting in a less substantial mitigation of membrane fouling. A similar variation trend was found in the increase in the Y value at the end of nanofiltration (Appendix A). Under crossflow filtration, the formation of a fouling layer is a dynamic equilibrium process. The foulants on the feed side migrate to the membrane surface driven by TMP, and the foulants deposited on the membrane surface also migrate to the feed side by water scouring [52]. The disturbance of the fluid to the foulants on the membrane surface was enhanced by the increased crossflow velocity, thereby promoting the migration of foulants from the membrane surface to the feed side and mitigating membrane fouling. Moreover, the mitigation of the permeance attenuation of the crossflow velocity increasing from 7 to 10 cm/s was much weaker than that increasing from 4 to 7 cm/s, while the energy consumption was the same, suggesting that an increase in crossflow velocity from 4 to 7 cm/s was appropriate for membrane-fouling control.

##### Solute Retention

The removal of COD, TDS, and total hardness by the MPCM2 was improved with an increase in crossflow velocity, as shown in Figure 4f. The removal of these indices for MPCM2 nanofiltration improved with an increase in crossflow velocity. Taking COD as an example, the removal efficiency at a crossflow velocity of 4 cm/s was 91.56%, which increased to 94.87% as the crossflow velocity increased to 10 cm/s. This was because the turbulence of the water near the membrane surface was enhanced by an increased crossflow velocity [52], which reduced the concentration polarization near the membrane surface. In other words, fewer foulants were deposited on the membrane surface, and its rejection by MPCM2 was enhanced by the increased crossflow velocity. Therefore, increasing the crossflow velocity was beneficial for improving the effluent quality. However, the changes in the removal efficiencies of COD, TDS, and total hardness by MPCM2 with the crossflow velocity were not substantial. Considering the permeance attenuation (Figure 4e), Y values (Appendix A), pollutant removal (Figure 4f), and energy consumption, a crossflow velocity of 7 cm/s was chosen for subsequent experiments.

### 3.3. Nanofiltration Performance and Membrane Fouling Analysis at Optimal Conditions

#### 3.3.1. Nanofiltration Performance

As discussed above, MPCM2 and operating conditions of 0.5 MPa, 25 °C, and 7 cm/s were chosen for subsequent experiments. A comprehensive analysis of the nanofiltration performance was conducted, and the results are listed in Appendix A. It was found that the values for Na^+^, Fe^3+^, Cl^–^, SO_4_^2–^, COD, total hardness, TDS, and turbidity of brackish water exceeded the standard values, and all the parameters met the hygienic standard for drinking water after nanofiltration by MPCM2, suggesting that MPCM2 is efficient in treating brackish water.

#### 3.3.2. Modeling of Membrane Fouling

The fitting curves of the membrane permeance under the optimal operating conditions based on the membrane fouling models are shown in Figure 5, and the relevant fitting parameters are listed in Appendix A. Of the four membrane fouling models, the cake filtration model could best fit the experimental data, with the largest R^2^ value of 0.93947 and lowest SSE value of 0.00515. This indicates that cake layer fouling was the main fouling mechanism for brackish water treatment using MPCM2.

#### 3.3.3. XPS Analysis

The XPS analysis results for virgin and fouled MPCM2 are listed in Table 1. The virgin membrane consisted of four elements, C, N, O, and S. Of these, S originated from the PS membrane, N from the modified polyamide layer, and both the PS membrane and modified polyamide layer contained C and O. After nanofiltration, the relative contents of C and S increased, and Cl, Na, Ca, and Mg were found on the fouled MPCM2, which were not present in virgin MPCM2. The increase in the contents of C and S might be related to the deposition of organic matter and carbonate/sulfate precipitate on the membrane fouling layer. The appearance of Ca might have resulted from the formation and deposition of CaSO_4_ on the membrane surface. Moreover, Ca^2+^ has been widely reported to act as a bridge between organic matter by binding with the organic carboxyl groups [53,54], which may also occur during the nanofiltration process. Therefore, the appearance of Ca in fouled MPCM2 might also be related to the interaction of Ca^2+^ with organic matter deposited on the membrane surface [55]. The appearance of Mg might be related to the formation of MgCO_3_ in the membrane fouling layer. These results suggest that organic and inorganic fouling were the primary fouling types for the nanofiltration of brackish water by MPCM2.

As shown in Figure 2, the CA of the virgin MPCM2 was 27.57°, suggesting excellent hydrophilicity of MPCM2. The CA of fouled MPCM2 increased to 57.76°, indicating that hydrophobic substances existed on the fouling layer and reduced its hydrophilicity. Inorganic salts and their sediments with strong polarity easily bound to the water molecules, whereas the CA value largely increased after the nanofiltration process. This indicated that hydrophobic organic matter existed in the fouling layer, and the continuously distributed foulants observed in the SEM image of the fouled MPCM2 (Figure 8e) could also illustrate this statement to a certain extent. Combining the results of the XPS analysis listed in Table 1, it could be said that the nanofiltration fouling by brackish water of MPCM2 was caused by the combined action of inorganic and organic fouling.

### 3.4. Determination of Cleaning Procedures for MPCM2 Fouling

#### 3.4.1. Determination of Cleaning Procedure

Though both membrane modification and changes in operating conditions can mitigate membrane fouling to a certain extent, membrane fouling still occurs, and membrane cleaning is needed. In this study, acid cleaning with HCl (pH = 2), alkali cleaning with NaOH (pH = 11), surfactant cleaning with 2% SDS (pH = 11), and complexing agent cleaning with EDTA (pH = 11) were used according to the actual fouling forms and properties of the MPCM2.

First, single-step cleaning experiments were conducted, and ultrapure water cleaning was conducted as a control to select the suitable cleaning agents for subsequent research on two-step cleaning (Figure 6). The cleaning efficiency was the lowest for ultrapure water cleaning with a *PR* of only 23.12%. Both acid and alkali treatments enhanced the cleaning efficiency. Acid and alkali cleaning corresponded to the dissolution of inorganic salt sediments and the dissolution or dispersion of organic matter, respectively. The *PR* for acid cleaning was 58.43%, while that for alkali was 67.54%, which further confirms the co-occurrence of inorganic and organic fouling during the MPCM2 nanofiltration of brackish water (Section 3.3). Furthermore, SDS and EDTA had higher cleaning efficiencies, and the *FR*s increased to 78.86 and 81.67%, respectively. SDS is an amphipathic molecule that is hydrophilic at one end and hydrophobic at the other. Its hydrophobic end can combine with organic matter through hydrophobic interactions, breaking away from MPCM2 under the disturbance of the SDS flow and achieving recovery of membrane performance. EDTA is a strong complexing agent. The high efficiency of EDTA cleaning suggests that the bridging action for Ca^2+^ might indeed exist between the membrane and organic matter, and Ca^2+^ tends to combine with EDTA owing to its stronger complexing effect [56]. Hence, the combination of organic matter with MPCM2 was weakened, and the organic matter broke away from the MPCM2 under the disturbance of the EDTA flow, which led to the highest cleaning efficiency among the five single-step cleaning methods.

To further increase the *FR*, acid cleaning (HCl, pH = 2) and complexing agent cleaning (EDTA, pH = 11) were chosen for two-step cleaning, with the aim of removing both inorganic and organic fouling. Two-step cleaning with different cleaning orders was conducted: EDTA cleaning followed by HCl cleaning (EDTA–HCl) and HCl cleaning followed by EDTA cleaning (HCL–EDTA). As shown in Figure 6, the cleaning efficiency of EDTA–HCl cleaning (*FR*: 98.77%) was higher than that of HCl–EDTA cleaning (*FR*: 86.54%), indicating that the cake layer and pore blocking formed on the MPCM2 were almost completely removed by EDTA–HCl cleaning. Moreover, the distinction in the cleaning efficiencies of the two cleaning orders might be related to the reaction between inorganic salt sediments and organic matter. Organic foulants are usually in a gel state, and inorganic salt sediments may be wrapped in gels, resulting in the less effective direct contact and reaction between the HCl agent and the inorganic salt sediments. This led to a much lower cleaning efficiency of HCl–EDTA cleaning compared to that of EDTA–HCl cleaning. 

#### 3.4.2. Influence of Membrane Cleaning on MPCM2 Properties

The *PR* after membrane cleaning may reflect the cleaning efficiency to a certain extent, but more aspects need to be considered for a more comprehensive examination of cleaning efficiency. In this study, the rejection rates of PEG200/Na_2_SO_4_ and CA, as well as the AFM images of the virgin, fouled, and cleaned MPCM2 were tested. The results are shown in Figure 7 and Figure 8. The rejection of the neutral solute by the nanofiltration membrane was based on the aperture-screening mechanism, and PEG200 with a concentration of 200 mg/L was chosen as the targeted neutral solute to estimate the variation in the membrane. Compared to the virgin MPCM2, the rejection rate of PEG200 by fouled MPCM2 increased from 78.57 to 81.43%, and that for cleaned MPCM2 decreased to 79.02%. This suggested that foulants deposited on the membrane surface blocked the membrane aperture to a certain extent. Moreover, the removal efficiencies of PEG200 for virgin and cleaned MPCM2 were fundamentally the same, indicating that the identified cleaning procedure did not damage the aperture structure. 

Na_2_SO_4_ removal was tested to investigate the influence of cleaning on desalination performance, and the Na_2_SO_4_ concentration was set to 1000 mg/L. As shown in Figure 7, the rejection rates of Na_2_SO_4_ by virgin, fouled, and cleaned MPCM2 were 96.46, 95.57 and 96.89%, respectively. The decrease in the rejection rates of Na_2_SO_4_ might be related to the decrease in the negative charge density of the surface of the membrane after nanofiltration, which was caused by the deposition of positive ions on the membrane surface as proved by the XPS analysis (Table 1). Moreover, the rejection rates of Na_2_SO_4_ by the virgin and cleaned MPCM2 were almost identical, indicating that the identified cleaning procedure did not mitigate the desalination performance. As discussed in Section 3.3, hydrophobic substances existed on the fouling layer, and the hydrophilicity of fouled MPCM2 was lower than that of virgin MPCM2. The CA of cleaned MPCM2 returned to its original level of virgin MPCM2, which suggested that foulants deposited on the MPCM2 surface were eliminated by EDTA–HCl cleaning, and its hydrophilic property was recovered.

Figure 8 shows the AFM and SEM images of the virgin, fouled, and cleaned MPCM2. A distinct concave-convex structure and membrane pores were observed on the virgin MPCM2 surface, while it was covered by foulants after nanofiltration, thus decreasing the surface roughness [54]. The foulants on the MPCM2 surface were effectively removed by EDTA–HCl cleaning and the concave-convex structure and membrane pores of MPCM2 were recovered. The morphological features of the cleaned MPCM2 were almost the same as those of virgin MPCM2, suggesting that the intrinsic structure of MPCM2 was mostly unchanged. This again confirms that an optimized cleaning procedure could effectively remove the foulants on MPCM2 surface and recover its properties.

## 4. Conclusions

This study established an efficient and high-performance nanofiltration system for the treatment of brackish water in northwest China. The modification using TEOA effectively enhanced the anti-fouling performance of PCM, and the MPCM2 modified with 2% TEOA was identified as the optimum membrane. The optimal operating conditions were a TMP of 0.5 MPa, a temperature of 25 °C, and a crossflow velocity of 7 cm/s based on the principle of low-carbon economy. Under the optimal operating conditions, cake layer fouling was the main form of membrane fouling for brackish water treatment, and membrane fouling was a combination of inorganic and organic fouling. Moreover, a membrane cleaning procedure of EDTA cleaning followed by HCl cleaning with a *PR* of 98.77% was identified as the optimal cleaning procedure and had almost no effect on the properties of MPCM2.

## Figures and Tables

**Figure 1 membranes-13-00038-f001:**
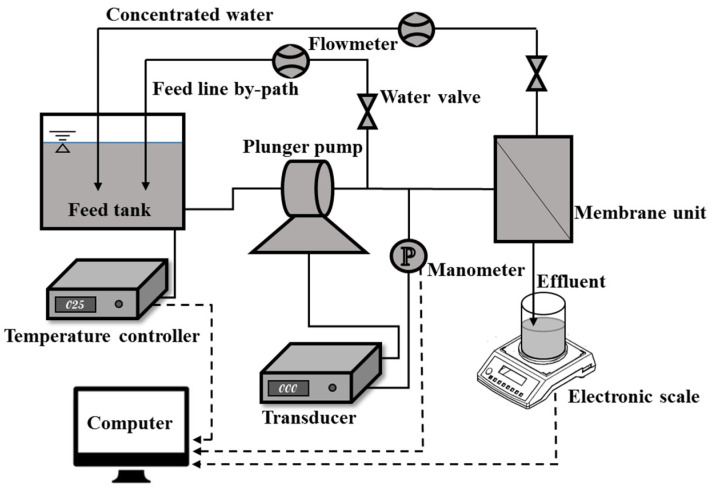
Continuous flow nanofiltration set-up.

**Figure 2 membranes-13-00038-f002:**
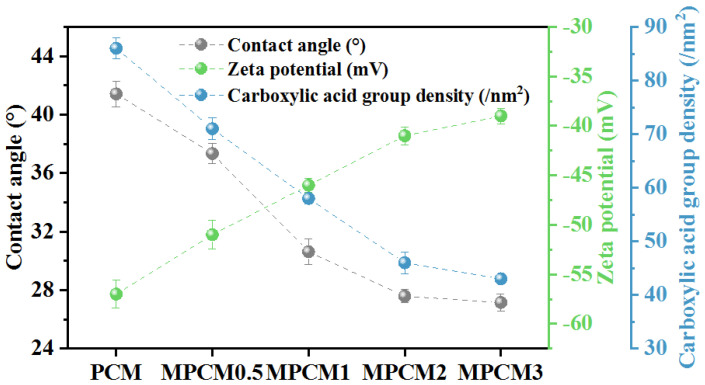
Physiochemical characteristics of the PCMs and MPCMs.

**Figure 3 membranes-13-00038-f003:**
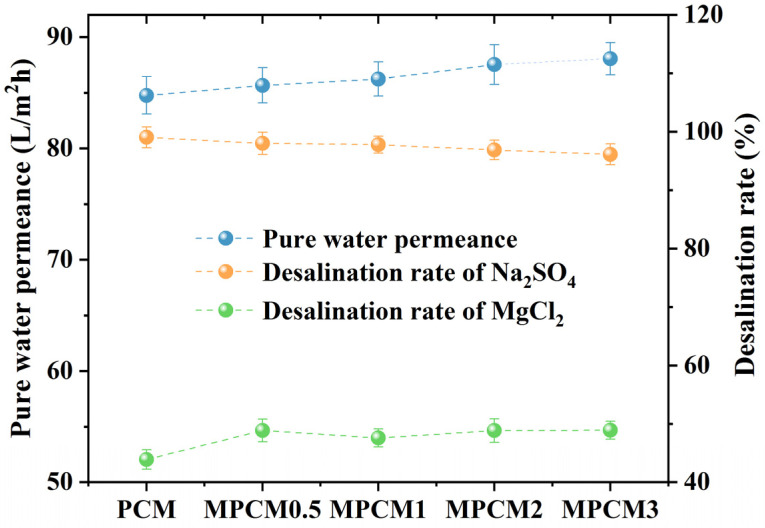
Nanofiltration performances of the PCMs and MPCMs.

**Figure 4 membranes-13-00038-f004:**
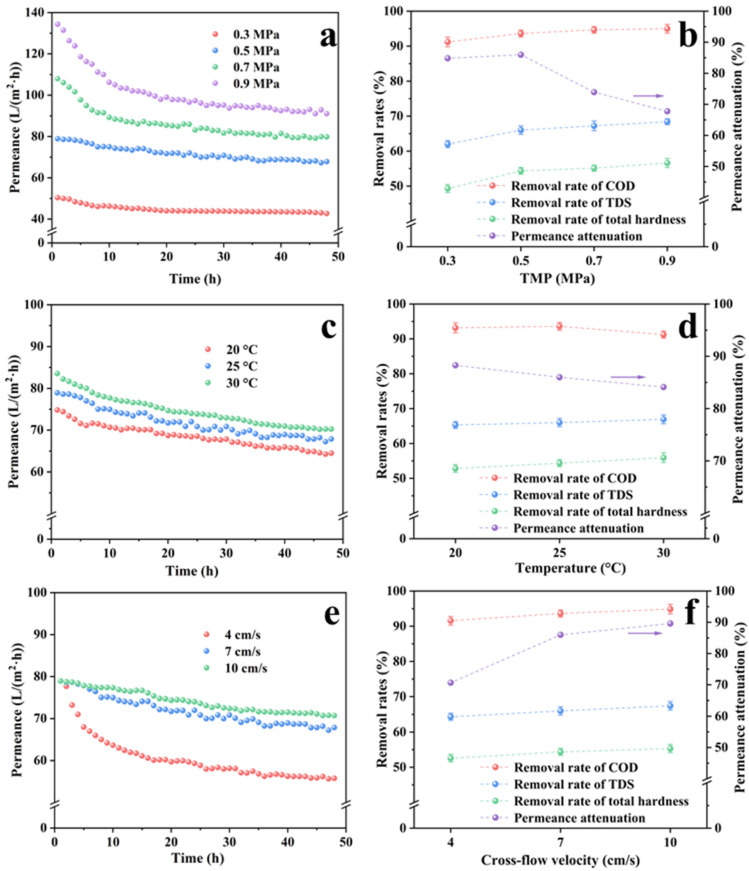
Influence of operating conditions on membrane permeance and nanofiltration performance of MPCM2: (**a**,**b**) TMP; (**c**,**d**) temperature; and (**e**,**f**) crossflow velocity.

**Figure 5 membranes-13-00038-f005:**
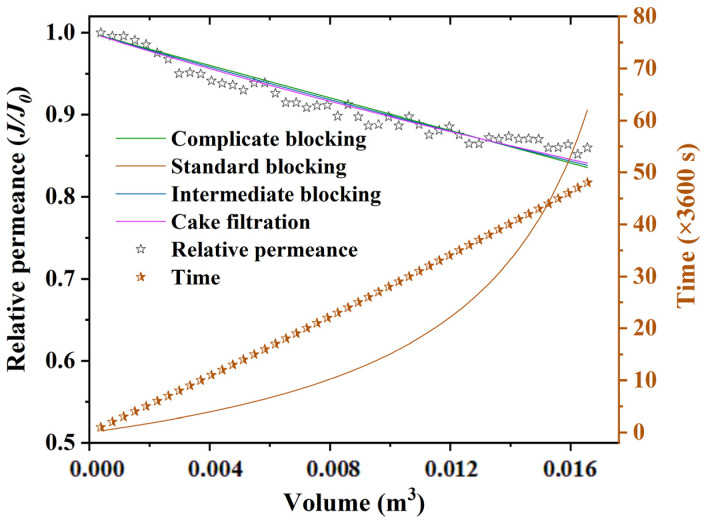
Fitting curves of membrane permeance under the optimal operating conditions based on membrane fouling models.

**Figure 6 membranes-13-00038-f006:**
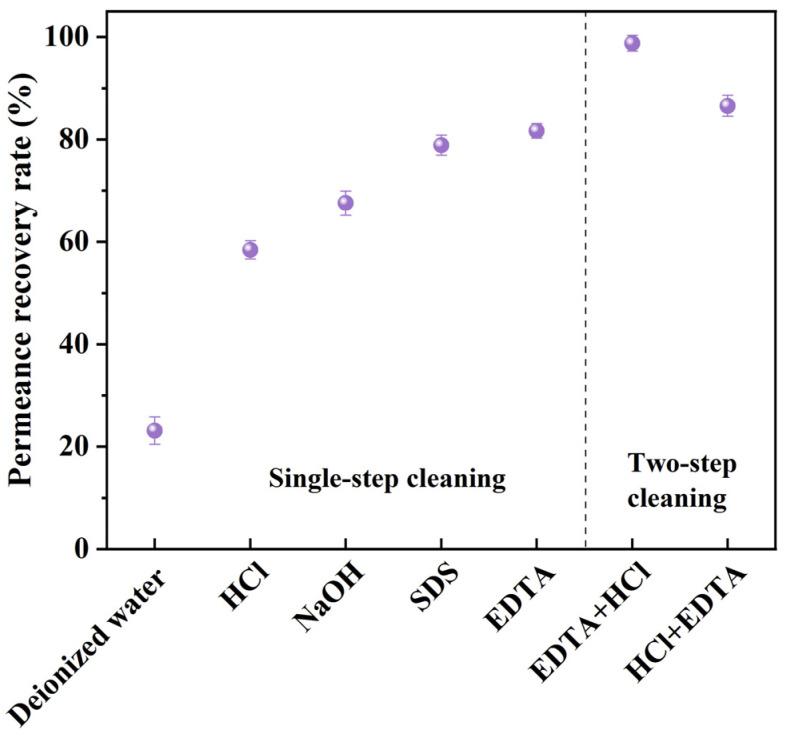
Permeance recovery rates after different cleaning procedures.

**Figure 7 membranes-13-00038-f007:**
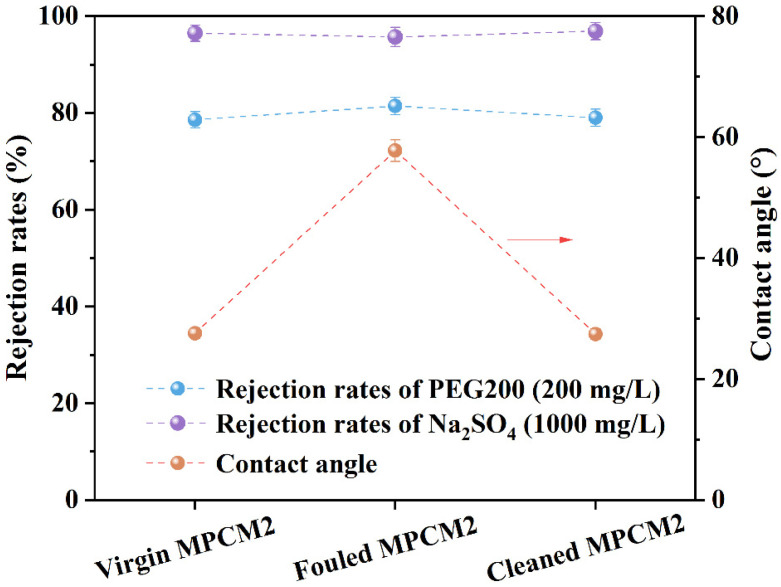
Influence of EDTA–HCl cleaning on MPCM2 properties.

**Figure 8 membranes-13-00038-f008:**
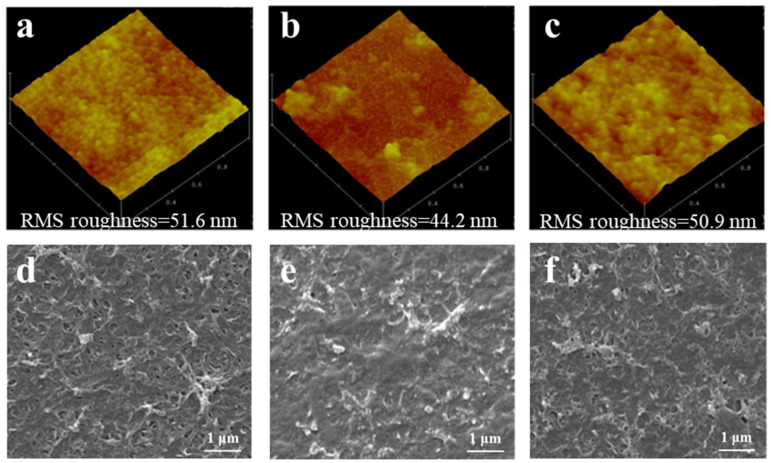
AFM images of (**a**) virgin, (**b**) fouled, and (**c**) cleaned MPCM2; SEM images of (**d**) virgin, (**e**) fouled, and (**f**) cleaned MPCM2.

**Table 1 membranes-13-00038-t001:** Element compositions of the virgin and fouled membranes based on XPS analysis.

Element	Contents (%)		Element	Contents (%)	
	Virgin MPCM2	Fouled MPCM2		Virgin MPCM2	Fouled MPCM2
C	67.22	68.43	Cl	0.00	0.02
N	10.57	9.16	Na	0.00	0.04
O	12.62	11.64	K	0.00	0.00
S	9.59	9.86	Mg	0.00	0.26
Ca	0.00	0.59			

## Data Availability

The data presented in this study are available in this article.

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
