# Peer review of "Evaluation of Membrane Fouling Control for Brackish Water Treatment Using a Modified Polyamide Composite Nanofiltration Membrane"

_membranes, 2022, doi:10.3390/membranes13010038_

Round 1
Reviewer 1 Report
This study focuses on NF membranes for brackish water treatment. The fouling behavior was investigated, and the cleaning methods were optimized to achieve long-term operation. The results are of interest to the community. My other comments are shown below.
1. Please use permeance instead of flux in all of the figures and discussions.
2. Figures should be re-drawn to improve clarity. For example, Figure 2 has three Y-axis’s, which is very confusing. It would be better to separate it into two sub-figures. Figure 5 is impossible to understand.
3. Labels in the figures can be made right by curves or points.
4. Please discuss the importance of the TEOA in the Conclusion.
5. What are the foulants in brackish water?
6. Please compare the lab-made membranes with commercial NF membranes. Why did the authors not study commercial ones as baselines?
Reviewer 2 Report
This manuscript analyzes the effect of membrane modification with triethanolamine (TEOA) and operating conditions on the nanofiltration effluent flux. In addition, the authors suggested some guidelines for cleaning fouled membranes. The manuscript is well-organized and includes many interesting results.
Meanwhile, as a reviewer, my comments about this manuscript are as follows:
1- Since you have analyzed the effect of many features on the nanofiltration behavior, I think it was better to use a design of experiments. Please explain.
2- It is necessary to justify whether the temperature, TMP, time, and cross-flow velocity ranges are covered the real-field situations or not?
3- Why the rejection rate of the virgin, fouled, and cleaned media are almost equal in Fig. 7?
4- How did you find the optimum values of the membrane and operating conditions?
5- You have many interesting results to provide in the conclusion section. Please add more key findings to this section.
Reviewer 3 Report
This work investigated the fouling propensity of the TEOA modified NF membrane in brackish water desalination under various operating conditions. A two-step cleaning procedure was developed to achieve membrane cleaning efficacy of 98.8% in terms of flux recovery. The cleaning part of the study is very well designed. The comments are as follows:
Major comments:
1. The novelty of the study needs to be emphasized. The statement in the introduction (Lines 65-67) is not very clear. It is noted that the findings of how the operating conditions (i.e., TMP, temperature, and crossflow velocity) impact membrane performance are not new. These were reported decades ago. How combining the membrane modification with optimization of operating conditions make this present work novel needs to be stated clearly. Did authors see a different trend with the modified membrane compared to unmodified commercial membrane? Is there any significant performance improvement than the commercial NF membranes by combining the membrane surface modification and operating condition optimization? If so, by how much?
2. The results of zeta potential are also confusing. The modified membrane had a decreased negative charge density, and why is that desired? How does membrane zeta potential impact membrane fouling or cleaning? Why decreased electro-negativity of the modified membrane led to decreased Na2SO4 rejection but increased MgCl2 rejection?
Minor comments:
1. Some terminologies are confusing. By desalination efficiency and removal efficiency, do authors mean solute rejection?
2. The explanation of the results will be stronger if the actual CP value can be calculated and compared for each condition (i.e., TMP and crossflow velocity).
3. Need to report the system recovery (Y), which is the ratio between permeate and feed flowrates.
4. Contact angle itself cannot indicate surface hydrophilicity. Will need to calculate surface tension.
5. Membrane resistance and fouling resistance should be calculated to support the statement in Line 230.
6. The explanation of why crossflow velocity of 7 cm/s was selected over higher velocity (Lines 308-310) in not convincing. Is the energy consumption of 10 cm/s that significant compared to 7 cm/s even if higher crossflow velocity leads to improved membrane rejection and reduced fouling?
7. The fouled MPCM2 membrane demonstrated higher rejection for neutral solute but lower rejection for Na2SO4 is not explained. Might worthy looking at the zeta potential of the fouled membrane.
8. For the AFM images, need to report surface roughness and explain why membrane surface is smoother after fouling. Is it consistent with the literature?
9. Membrane surface characterization using SEM for the fouled membrane could be helpful in determining the potential foulant that deposited on the membrane surface.
Round 2
Reviewer 1 Report
The authors have addressed most of my comments. Figures 2 and 5 are still very difficult to follow. Please separate into 2 or more figures.
Reviewer 2 Report
The authors sufficiently addressed my comments. It seems the manuscript is now acceptable for the publicstion.
Author Response
Response: Thank you for the affirmation comments.
Reviewer 3 Report
The revisions of the manuscript are acceptable
Author Response

(The authors gave the same response as above.)
